# SENSE: SEMANTICALLY ENHANCED NODE SEQUENCE EMBEDDING

## ABSTRACT

Effectively capturing graph node sequences in the form of vector embeddings is critical to many applications. We achieve this by (i) first learning vector embeddings of single graph nodes and (ii) then composing them to compactly represent node sequences. Specifically, we propose SENSE-S (Semantically Enhanced Node Sequence Embedding - for Single nodes), a skip-gram based novel embedding mechanism, for single graph nodes that co-learns graph structure as well as their textual descriptions. We demonstrate that SENSE-S vectors increase the accuracy of multi-label classification tasks by up to $50\%$ and link-prediction tasks by up to $78\%$ under a variety of scenarios using real datasets. Based on SENSE-S, we next propose generic SENSE to compute composite vectors that represent a sequence of nodes, where preserving the node order is important. We prove that this approach is efficient in embedding node sequences, and our experiments on real data confirm its high accuracy in node order decoding.

## 1 INTRODUCTION

Accurately learning vector embeddings for a sequence of nodes in a graph is critical to many scenarios, e.g., a set of Web pages regarding one specific topic that are linked together. Such a task is challenging as: (i) the embeddings may have to capture graph structure along with any available textual descriptions of the nodes, and moreover, (ii) nodes of interest may be associated with a specific order. For instance, (i) for a set of Wikipedia pages w.r.t. a topic, there exists a recommended reading sequence; (ii) an application may consist of a set of services/functions, which must be executed in a particular order (workflow composability); (iii) in source routing, the sender of a packet on the Internet specifies the path that the packet takes through the network or (iv) the general representation of any path in a graph or a network, e.g., shortest path. Node sequence embedding, thus, requires us to (i) learn embeddings for each individual node of the graph and (ii) compose them together to represent their sequences. To learn the right representation of individual nodes and also their sequences, we need to understand how these nodes are correlated with each other both functionally and structurally.

A lot of work has only gone into learning single node embeddings (i.e., where node sequence length is 1), as they are essential in feature representations for applications like multi-label classification or link prediction. For instance, algorithms in Perozzi et al. (2014), Grover & Leskovec (2016), Tang et al. (2015) and others try to extract features purely from the underlying graph structure; algorithms in Le & Mikolov (2014), Mikolov et al. (2013) and others learn vector representations of documents sharing a common vocabulary set. However, many applications would potentially benefit from representations that are able to capture both textual descriptions and the underlying graph structure simultaneously. For example, (1) classification of nodes in a network not only depends on their inter-connections (i.e., graph structure), but also nodes' intrinsic properties (i.e., their textual descriptions); (2) for product recommendations, if the product is new, it may not have many edges since not many users have interacted with it; however, using the textual descriptions along with the graph structure allows for efficient bootstrapping of the recommendation service. For general case of sequence lengths greater than 1, despite the importance in applications like workflow composability described above, there is generally a lack of efficient solutions. Intuitively, we can concatenate or add all involved node vectors; however, such a mechanism either takes too much space or loses the sequence information; thus unable to represent node sequences properly.

We aim to learn node sequence embeddings by first first addressing the single node embedding issue, as a special case of node sequence embedding, by considering both the textual descriptions and the graph structure. We seek to answer two questions: How should we combine these two objectives? What framework should we use for feature learning? Works that jointly address these two questions either investigate them under different problem settings (Cao et al., 2017; Xiao et al., 2017), under restricted learning models (Le & Lauw, 2014), ignore the word context within the document (Liao et al., 2017; Wang et al., 2017), do not co-learn text and graph patterns (Li et al., 2016) or only consider linear combinations of text and graph (Garcia Duran & Niepert, 2017); this is elaborated further in Section 2. In contrast, we propose a generic neural-network-based model called SENSE-S (Semantically Enhanced Node Sequence Embeddings - for Single nodes) for computing vector representations of nodes with additional semantic information in a graph. SENSE-S is built on the foundation of skip-gram models. However, SENSE-S is significantly different from classic skip-gram models in the following aspects: (i) For each word $\phi$ in the textual description of node $v$ in the given graph, neighboring words of $\phi$ within $v$'s textual description and neighboring nodes of $v$ within the graph are sampled at the same time. (ii) The text and graph inputs are both reflected in the output layer in the form of probabilities of co-occurrence (in graph or text). (iii) Moreover, this joint optimization problem offers an opportunity to leverage the synergy between the graph and text inputs to ensure faster convergence. We evaluate the generated vectors on (i) Wikispeedia (2009) to show that our SENSE-S model improves multi-label classification accuracy by up to $50\%$ and (ii) Physics Citation dataset (Leskovec & Krevl, 2014) to show that SENSE-S improves link prediction accuracy by up to $78\%$ over the state-of-the-art.

Next, we propose SENSE for general feature representation of a *sequence* of nodes. This problem is more challenging in that (i) besides the original objectives in SENSE-S, we now face another representation goal, i.e., sequence representation while preserving the node order; (ii) it is important to represent the sequence in a compact manner; and (iii) more importantly, given a sequence vector, we need to be able to decipher which functional nodes are involved and in what order. To this end, we develop efficient schemes to combine individual vectors into complex sequence vectors that address all of the above challenges. The key technique we use here is vector cyclic shifting, and *we prove that the different shifted vectors are orthogonal with high probability*. This sequence embedding method is also evaluated on the Wikispeedia and Physics Citation datasets, and the accuracy of decoding a node sequence is shown to be close to $100\%$ when the vector dimension is large.

## 2 RELATED WORK

We overview the most related works by categorizing them as follows:

**Learning vector representation from text:** Vector representation of words (Schütze, 1993) has been a long standing research topic. It has received significant attention in the recent times due to the advances in deep neural networks (Bengio et al., 2003; Mikolov et al., 2009; 2013). In particular, these neural-network-based schemes outperform $n$-gram-based techniques (Katz, 1987; Jelinek & Mercer, 1980) significantly as they are able to learn the similarities between words. Furthermore, paragraph2vec (Le & Mikolov, 2014) extends the well-established word2vec (Mikolov et al., 2013) to learn representations of chunks of text.

**Learning vector representation from graphs:** Lot of research has gone into learning graph representations by translating the network/graph into a set of words or documents (Perozzi et al., 2014; Pimentel et al., 2017; Ribeiro et al., 2017; Wang et al., 2016; Liu et al., 2016; Tang et al., 2015; Grover & Leskovec, 2016). Generic models incorporating both edge weight and direction information for graph embeddings are proposed in Zhou et al. (2017), Tang et al. (2015) and Grover & Leskovec (2016). Specifically, node2vec (Grover & Leskovec, 2016) advances the state-of-the-art in this area by designing flexible node sampling methodologies to allow feature vectors to exhibit different properties. Subgraph2vec (Narayanan et al., 2016) extends these schemes to learn vector representations of subgraphs. Simpkin et al. (2018) proposes techniques to represent graph sequences under the assumption that each node is represented by a random binary vector.

**Learning graph representation with auxiliary information:** Broadly speaking, our work falls into the category of node embedding in graphs with auxiliary information. Guo et al. (2017), Xie et al. (2016), Huang et al. (2017a) and others address the case where nodes are associated with labels. Niepert et al. (2016) studies graph embedding when node/edge attributes are continuous. Cao et al.

(2017) investigates phrase ambiguity resolution via leveraging hyperlinks. However, all these works operate under information or network constraints. On the other hand, Xiao et al. (2017), Yao et al. (2017) and Wang & Li (2016) explore embedding strategies in the context of knowledge graphs, where the main goal is to maintain the entity relationships specified by semantic edges. In contrast, we consider a simpler network setting where only nodes are associated with semantic information. EP (Garcia Duran & Niepert, 2017) and GraphSAGE (Hamilton et al., 2017) learn embeddings for structured graph data. However, the textual similarities are only captured by linear combinations. Planetoid (Yang et al., 2016) computes node embeddings under semi-supervised settings; metap-ath2vec (Dong et al., 2017) learns embeddings for heterogeneous networks (node can be author or paper); and Graph-Neural-Network-based embeddings are explored in Kipf & Welling (2017) and Li et al. (2016). However, these papers do not explicitly learn graph structure and text patterns simulta-neously, and thus are complementary to SENSE. SNE (Liao et al., 2017), SNEA (Wang et al., 2017), TADW (Yang et al., 2015), HSCA (Zhang et al., 2016), AANE (Huang et al., 2017b), ANRL (Zhang et al., 2018) and PLANE (Le & Lauw, 2014) are more related to our work. However, unlike SENSE, these do not consider the relative context of the words w.r.t. the document. Furthermore, the ob-jective of PLANE is to maximize the likelihood that neighboring nodes have similar embeddings, which is not always the case in practice because neighboring nodes may be semantically different; more critically, it relies on strong assumptions of statistical distributions of words and edges in the network. In this regard, we propose a generic embedding scheme that jointly considers network topology as well as the nodes' semantic information.

## 3 SENSE-S: SENSE FOR SINGLE NODE EMBEDDINGS

To embed a general node sequence, we first consider a special case where each node sequence contains only one node. Such single node embedding is referred to as SENSE-S, which jointly learns node representations along with textual descriptions in graphs.

### 3.1 SENSE-S OBJECTIVE

Let $\mathcal{G} = (V, E)$ denote a given directed or undirected graph, where $V$ is the set of nodes and $E$ the set of edges. Each node in $V$ is associated with a text description. We aim to embed each node $v$ in $V$ into a feature vector that captures both graphical (neighboring node inter-connections) and textual (semantic meaning of $v$) properties. Specifically, let $\phi$ denote a word in the text description of node $v$. Suppose we obtain a set of neighboring nodes of $v$ in graph $\mathcal{G}$ via a specific node sampling strategy, e.g., biased random walk (Grover & Leskovec, 2016), and a set of neighboring words of $\phi$ in the text description of $v$ by a sliding window over consecutive words (Mikolov et al., 2013). We then define $N_{\mathcal{G}}(v)$ as a probabilistic event of observing the set of neighboring nodes of $v$ in $\mathcal{G}$ (under the chosen model) and $N_T(\phi|v)$ as an event of observing the set of neighboring words of $\phi$ in the text description of $v$. Let $f : v \rightarrow \mathbb{R}^d$ ($v \in V$) be the embedding function that maps each node $v$ in $V$ into a $d$-dimensional vector. Our goal is to find function $f$ that maximizes:

$$\max_f \sum_{v \in V} \sum_{\phi \text{ in } v} \log \Pr[N_{\mathcal{G}}(v) N_T(\phi|v)|\phi, f(v)]. \tag{1}$$

Since events $N_{\mathcal{G}}(v)$ and $N_T(\phi|v)$ are independent, (1) can be rewritten as:

$$\max_f \sum_{v \in V} w_v \log \Pr[N_{\mathcal{G}}(v)|f(v)] + \sum_{v \in V} \sum_{\phi \text{ in } v} \log \Pr[N_T(\phi|v)|\phi, f(v)], \tag{2}$$

where $w_v$ is the number of words in the description of $v$. Given a word in a node, (2) jointly captures the node neighborhood in the graph and the word neighborhood in text.

### 3.2 SENSE-S ARCHITECTURE

We build SENSE-S on the foundation of the skip-gram model (Mikolov et al., 2013). In particular, Grover & Leskovec (2016) and Perozzi et al. (2014) leverage this skip-gram model to learn vector representation of nodes in a graph by performing biased random walks on the graph and treating each walk as equivalent to a sentence, aiming to predict a neighboring *node* given the current node in a

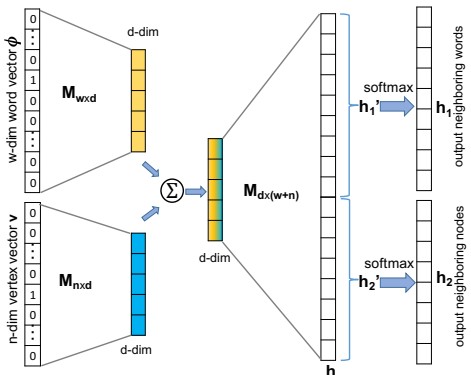 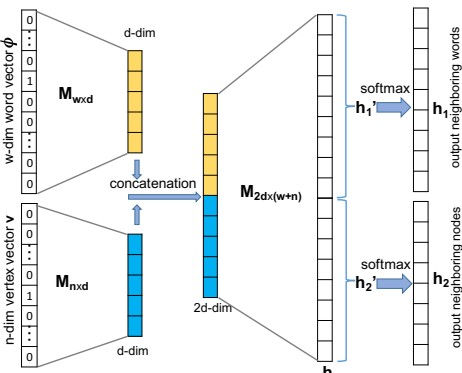

Figure 1: SENSE-S (add) architecture      Figure 2: SENSE-S (concat) architecture

graph. On the other hand, Le & Mikolov (2014) extends skip-gram models to learn embeddings of various chunks of text, e.g., sentences, paragraphs or entire documents, via sampling the neighboring words over a sliding window within the text. Motivated by the effectiveness of these models, we build two SENSE-S models, SENSE-S (add) and SENSE-S (concat), as detailed below.

**SENSE-S (add):** Let $w$ denote the number of words (some uninformative words are skipped) in text descriptions across *all* nodes in the given graph $\mathcal{G}$, i.e., $w$ is the size of the entire vocabulary, and $n$ the number of nodes in $\mathcal{G}$. Then our model is formulated as a neural network, as shown in Figure 1. In this model, each input is a two-tuple $(\phi, v)$, i.e., word $\phi$ is picked from the description of node $v$. Then $(\phi, v)$ is mapped to two one-hot vectors, $w$-dimensional word vector $\phi$ and $n$-dimensional vertex vector $\mathbf{v}$, where only the entries corresponding to $\phi$ and $v$ are set to 1 and others are set to 0. Then as in typical fully connected neural network architectures, in the first layer, $\phi^T$ and $\mathbf{v}^T$ are multiplied (implemented as look-up for efficiency) by two matrices $\mathbf{M}_{w \times d}$ and $\mathbf{M}_{n \times d}$, respectively. The resulting vectors are then added together and multiplied by another matrix $\mathbf{M}_{d \times (w+n)}$ in the second layer, i.e., a $(w + n)$-dimensional vector $\mathbf{h}^T := (\phi^T \mathbf{M}_{w \times d} + \mathbf{v}^T \mathbf{M}_{n \times d}) \mathbf{M}_{d \times (w+n)}$ is obtained. Finally, unlike typical skip-gram models where all entries in the output vectors of the second layer are processed by a softmax function, we decompose vector $\mathbf{h}$ into two sub-vectors, $\mathbf{h}_1'$ consisting of the first $w$ entries and $\mathbf{h}_2'$ consisting of the rest. Then $\mathbf{h}_1'$ and $\mathbf{h}_2'$ are fed to separate softmax functions, yielding $\mathbf{h}_1$ and $\mathbf{h}_2$, respectively (see Figure 1). The reason for this decomposition operation is that we use $\mathbf{h}_1$ to represent the probability vector of neighboring words of $\phi$ in the description of node $v$, and $\mathbf{h}_2$ to represent the probability vector of neighboring nodes of $v$ in the graph. Using this neural network architecture, we aim to learn the values of all entries in the matrices $\mathbf{M}_{w \times d}$, $\mathbf{M}_{n \times d}$ and $\mathbf{M}_{d \times (w+n)}$ such that the entries (i.e., probabilities) in $\mathbf{h}_1$ and $\mathbf{h}_2$ corresponding to neighboring words of $\phi$ (within the text description of node $v$) and neighboring nodes of $v$ (within the given graph) are maximized; see the objective in (2). We use matrix $\mathbf{M}_{n \times d}$ as our final semantically augmented node embeddings, i.e., each row in $\mathbf{M}_{n \times d}$ corresponds to a node embedding vector.

Note that a unique property of SENSE-S is that it is a conjoined model where the textual and graphical inputs both contribute to the learning of node embeddings. Moreover, there is an *add* operation for combining these features, and thus this model is called SENSE-S (add). This is in contrast to the *concatenation* operation that is used in a different implementation of SENSE-S; see below.

**SENSE-S (concat):** SENSE-S (concat) model is illustrated in Figure 2. Clearly, SENSE-S (concat) is quite similar to SENSE-S (add), except that (i) the resulting vectors generated by the first layer are *concatenated*, and (ii) the matrix dimension in the second layer becomes $(2d) \times (w + n)$.

## 4 SENSE

In SENSE-S, our focus has been on computing semantically enhanced embeddings for individual nodes. In this section, we propose the general SENSE to represent any set of nodes following a specified order using the node vectors generated by SENSE-S, called *node sequence embedding*.

Given the original graph $\mathcal{G} = (V, E)$, let $S = v_1 \rightarrow v_2 \rightarrow \cdots \rightarrow v_q$ be a node sequence constructed with $v_i \in V(i = 1, 2, \ldots, q)$. Note that $S$ may contain *repeated* nodes, e.g., some functions need

to be executed more than once in one application. Intuitively, node sequence $S$ can be represented by a $d \times q$ matrix with column $i$ being the vector representation of node $v_i$. However, such a representation is costly in space. Alternatively, representing $S$ by the vector sum $\sum_{i=1}^{q} \mathbf{v}_i$ ($\mathbf{v}_i$ corresponds to $v_i$) results in missing the node order information. Hence, in this section, we seek to find a low-dimensional vector representation such that (i) node properties in the original network $\mathcal{G}$ and (ii) the node order in $S$ are both preserved.

## 4.1 NODE SEQUENCE EMBEDDING MECHANISM

In this section, all node vectors are *unit vectors* obtained by normalizing the node vectors generated by SENSE-S; this property is critical in our node sequence embedding method (see Section 4.2).

**Node Sequence Vector Construction:** Given a node sequence $S = v_1 \rightarrow v_2 \rightarrow \cdots \rightarrow v_q$ following in order from node $v_1$ to node $v_q$, let $\mathbf{v}_i$ be the unit node vector of node $v_i$. We first perform positional encoding, via cyclic shift function. Specifically, given vector $\mathbf{v}$ of dimension-$d$ and non-negative integer $m$, we define $\mathbf{v}^{(m)}$ as a vector obtained via cyclically shifting elements in $\mathbf{v}$ by $m' = (m \mod d)$ positions. Therefore, $\mathbf{v}_i^{(i-1)}$ represents a vector resulted from cyclically shifting $(i - 1 \mod d)$ positions of the node vector at position $i$ in $S$. Let $\mathbf{S}$ denote the vector representation of node sequence $S$. Suppose $q \ll d$, Then we embed $S$ as $\mathbf{S} = \sum_{i=1}^{q} \mathbf{v}_i^{(i-1)}$. Note that for repeated nodes in $S$, they are cyclically shifted by different positions depending on the specific order in the node sequence. In $\mathbf{S}$, by imposing the restriction $q \ll d$, we ensure that wraparounds do not occur while shifting the vector, because it may lead to ambiguity of node positions within a node sequence. Simple as this embedding approach may seem, we show that it exhibits the following advantages. First, the dimension of node sequence vectors remains the same as that of the original node vectors. Second, given a node sequence vector $\mathbf{S}$, we are able to infer which nodes are involved in $\mathbf{S}$ and their exact positions in $\mathbf{S}$ as explained below.

**Node Sequence Vector Decoding:** The method for determining which nodes are included (and in which order) in a given node sequence vector is referred to as *node sequence vector decoding*. The basic idea in node sequence vector decoding is rooted in Theorem 2 (Section 4.2), which implies that using cyclic shifting, we essentially enable a preferable property that $\mathbf{v}_i^{(i-1)}$ and $\mathbf{v}_j^{(j-1)}$ with $i \neq j$ are almost orthogonal, i.e., $\mathbf{v}_i^{(i-1)} \cdot \mathbf{v}_j^{(j-1)} \approx 0$, even if $\mathbf{v}_i$ and $\mathbf{v}_j$ are related in the original graph ($\mathbf{v}_i$ and $\mathbf{v}_j$ may even correspond to the same node). By this property, we make the following claim, assuming that all node vectors are *unit vectors*.

**Claim 1.** *Given a node sequence vector* $\mathbf{S}$*, node* $v$*, whose node vector is* $\mathbf{v}$*, is at the $k$-th position of this node sequence if the inner product* $\mathbf{S} \cdot \mathbf{v}^{(k-1)} \approx 1$*.*

Claim 1 provides an efficient way to decode a given node sequence. In particular, to determine whether (and where) a node resides in a node sequence, it only takes quadratic complexity $O(d^2)$.

**Example:** Suppose we have a node sequence $S = ① \rightarrow ② \rightarrow ③$ consisting of three nodes. By SENSE-S and our encoding method, we construct its node sequence vector as $\mathbf{S} = \mathbf{v}_1^{(0)} + \mathbf{v}_2^{(1)} + \mathbf{v}_3^{(2)}$. Then each node can compute the inner product of its cyclic shifted vector with $\mathbf{S}$. If the result is approximately 1, then its position in this node sequence is uniquely determined. For instance, $\mathbf{S} \cdot \mathbf{v}_2^{(1)} = \mathbf{v}_1^{(0)} \cdot \mathbf{v}_2^{(1)} + \mathbf{v}_2^{(1)} \cdot \mathbf{v}_2^{(1)} + \mathbf{v}_3^{(2)} \cdot \mathbf{v}_2^{(1)} \approx 0 + 1 + 0 = 1$ (see Theorem 2). Thus, given the encoded node sequence $\mathbf{S}$, we know node 2 is at the second position.

## 4.2 THEORETICAL FOUNDATION

We now present our theoretical results, to support Claim 1. Proofs are presented in the appendix.

**Theorem 2.** *Let* $\mathbf{x}$ *and* $\mathbf{y}$ *be* $N$*-dimensional unit vectors with i.i.d. (independent and identically distributed) vector elements. If* $\mathbf{x} \cdot \mathbf{y} = c$*, then* $\mathbb{E}[\mathbf{x} \cdot \mathbf{y}^{(m)} | \mathbf{x} \cdot \mathbf{y} = c] = 0$ *and* $\mathrm{Var}[\mathbf{x} \cdot \mathbf{y}^{(m)} | \mathbf{x} \cdot \mathbf{y} = c] = (1 - c^2)/(N - 1)$*, where* $c$ *($|c| \leq 1$) is a constant and* $m \neq kN$ *($m, k \in \mathbb{Z}^+$).*

Theorem 2 suggests that when two nodes $a$ and $b$ in a graph are related, then their embedded vectors $\mathbf{v}_a \cdot \mathbf{v}_b = c$, where $c$ is non-zero. Nevertheless, if one of $\mathbf{v}_a$ and $\mathbf{v}_b$, say $\mathbf{v}_b$, is cyclically shifted, yielding $\mathbf{v}_b^{(m)}$ with $\mathbf{v}_b \neq \mathbf{v}_b^{(m)}$, then $\mathbf{v}_a \cdot \mathbf{v}_b^{(m)} = 0$ happens with high probability when the vector dimension is sufficiently large, although $\mathbf{v}_a$ and $\mathbf{v}_b$ are originally related.

# 5 EVALUATION

This section evaluates SENSE under varying scenarios, with multiple datasets and applications.

## 5.1 DATASETS

We evaluate SENSE on the following datasets that contain graph information along with textual descriptions: (1) Wikispeedia (2009): it contains both Wikipedia plain text articles (text descriptions) and hyper links between articles (graph structure). It is a directed graph with $4,604$ nodes (each is a Wikipedia article) and $119,882$ hyper links. (2) Citation Network (Leskovec & Krevl, 2014): it contains both textual descriptions (title, authors and abstract of the paper) and graphical structure (citations). It is a directed graph with $27,770$ nodes (papers) and $352,807$ edges (citations).

## 5.2 SENSE-S MODEL TRAINING

To train our SENSE-S model, we first define the loss function based on (2). Suppose the vocabulary size associated with each node $v \in V$ is similar, i.e., we assume $\forall v \in V, w_v \approx c$ ($c$ is a constant). Then let $F_{\mathcal{G}} := \sum_{v \in V} \log \Pr[N_{\mathcal{G}}(v)|f(v)]$ and $F_T := \sum_{v \in V} \sum_{\phi \text{ in } v} \log \Pr[N_T(\phi|v)|\phi, f(v)]$. We define the loss function as

$$L := -cF_{\mathcal{G}} - F_T. \tag{3}$$

We then use stochastic gradient descent to minimize our loss function. Let $\eta$ be the learning rate. At each iteration, we update the model parameters by adding a fraction of $-\nabla L$, i.e., $-\eta\nabla L = c\eta\nabla F_{\mathcal{G}} + \eta\nabla F_T$. Since the per-node vocabulary size is much larger than 1, the node neighborhood sampling via random walk is much less frequent than the textual neighborhood sampling. Therefore we want to inject more input data consisting of only the nodes' graphical neighborhood information. To do so, we adjust the model parameter update rule $-\eta\nabla L$ as

$$\beta_1 \nabla F_{\mathcal{G}} + \beta_2 \nabla F_T \ (\beta_1 > \beta_2 > 0), \tag{4}$$

where $\beta_1$ and $\beta_2$ are the equivalent learning rates for graph inputs and text inputs, respectively.

## 5.3 SENSE-S EVALUATION

We first evaluate SENSE-S, which is compared against the following baseline solutions.

**Leverage graph information alone:** To compare with schemes that use graph information alone, we use node2vec since it has been shown to be flexible to capture different graph properties.

**Leverage text information alone:** To compare against schemes that use textual information alone, we use semantic vectors from paragraph2vec since it outperforms other schemes such as Recursive Neural Tensor Networks (Socher et al. (2013)) for tasks like Sentiment Analysis. As in SENSE-S, we also study two implementations of paragraph2vec, i.e., addition and concatenation operations at the hidden layer, referred to as paragraph2vec (add) and paragraph2vec (concat), respectively.

**Leverage text+graph information:** For joint text/graph learning, we compare with the following:

1) **Initialize with semantic vectors:** We learn embeddings using node2vec, but rather than using random initialization, we initialze the vectors using paragraph2vec.

2) **Initialize with graphical vectors:** Here, we learn final embeddings using paragraph2vec, but initialize them with node2vec, i.e., just reverse of the scheme above.

3) **Iterative Vectorization:** The above approaches only leverage semantic or graphical vectors for initializations. Here, we try to capture both iteratively. Specifically, in one iteration, we compute node embedding via node2vec with the embeddings from the previous iteration as initializations; the corresponding results are then fed to paragraph2vec as initializations to further compute node embeddings, after which we go to the next iteration. We repeat this process multiple times (5 times in our experiment) to get the final embeddings.

4) **Concatenation of graphical and semantic vectors:** Here, we simply concatenate the vectors obtained from paragraph2vec and node2vec and use them as our node embedding vectors.

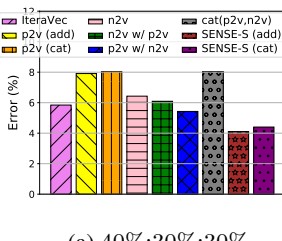 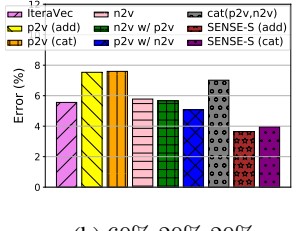 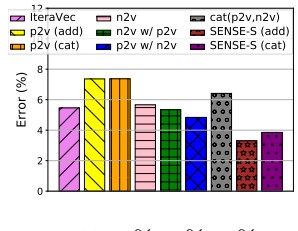

(a) 40%:30%:30%      (b) 60%:20%:20%      (c) 80%:10%:10%

Figure 3: Multi-label classification on Wikispeedia dataset for varying Train:Validation:Test splits of data. Document length = first 500 characters (n2v: node2vec, p2v (add)/(cat): paragraph2vec (add)/(concat), n2v w/ p2v: node2vec with paragraph2vec (add) initialization, p2v w/ n2v: paragraph2vec (add) with node2vec initialization, IteraVec: Iterative Vectorization, cat(p2v,n2v): concatenation of p2v and n2v vectors).

**Experimental Setup:** We first learn the vector representations and then use these vector representations for two different tasks: (i) *Multi-label classification:* Wikipedia pages are classified into different categories, such as history, science, people, etc. This ground truth information is included in the Wikispeedia dataset (which is not used while learning the vectors). There are 15 different top level categories, and our multi-label classification task tries to classify a page into *one or more* of these categories based on the vectors obtained from different algorithms. We train the OneVsRest-Classifier (SVM, linear kernel) from scikit-learn for this task. (ii) *Link prediction:* Since no category information is available for the Citation Network, we evaluate for link prediction. In particular, 1% of existing citations are removed, after which vectors are learned on this network. We use these removed links as positive samples for link prediction. For negative samples, we randomly sample the same number of pairs which are not linked via a citation in the original network. To obtain the *similarity features w.r.t. a pair of nodes*, after experimenting with several alternatives, we chose the element-wise absolute difference and train SVM classifier (linear kernel) for link prediction.

Parameter settings: (i) We perform $\kappa$ random walks starting from each node in the graph ($\kappa$ is 10 for Wikispeedia and 3 for Citation Network, since Citation Network is larger); (ii) each walk is of length 80 as recommended by Grover & Leskovec (2016); (iii) we use sliding window of size 5 for neighboring word sampling; (iv) the default node vector dimension is 128; (v) multi-label classification error is the misclassification rate over the test set; (vi) link prediction error is the percentage of incorrect link predictions over all pairs of papers in the test set; (vii) learning rates $\beta_1$ and $\beta_2$ in (4) are selected based on the validation set, and the error is reported on the test set.

**Multi-label classification error:** The error of multi-label classification is reported in Figure 3, where the first 500 characters of each Wikipedia page are selected as its textual description. Figures 3 (a), (b) and (c) correspond to different splits of the dataset (based on number of nodes) into training, validation and test samples. The following observations can be made. First, as expected, the results of all the schemes improve with higher ratio of training to test samples. Second, in general, schemes that leverage both textual and graphical information incur lower errors. Third, among the schemes that utilize both textual and graphical information, SENSE-S (add) and SENSE-S (concat) consistently perform the best. This is because we train the network to co-learn the textual and graphical information to ensure that both objectives converge. This is in contrast with other schemes where the two objectives, due to the loose coupling, are not guaranteed to converge. Finally, SENSE-S (add) (in Figure 3 (c)) outperforms node2vec by over 40%, paragraph2vec (add) by over 55% and the closest baseline scheme that leverages both text and graph by 30%. This confirms the benefit of co-learning features from textual as well as graphical information under the SENSE-S architecture. We also see similar trends using the first 1,000 characters and omit the results for brevity.

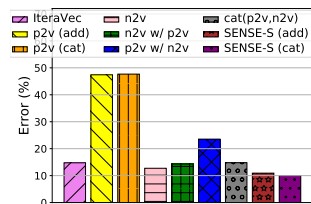

Figure 4: Link prediction in Citation Network.

**Link prediction error:** The error of link prediction in the Citation Network is reported in Figure 4. We fix train:valid:test to 60%:20%:20% (based on number of links) and use the first 500 characters as text description. We make several interesting observations. First, schemes that use graph information alone (node2vec) substantially outperform schemes that use text descriptions alone (paragraph2vec) for link prediction task. Intuitively, this is because the neighborhood information, which is important for link prediction task, is captured effectively by the node embeddings obtained from node2vec-like

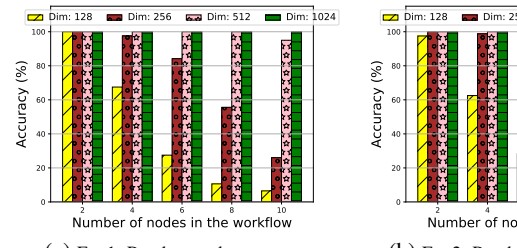
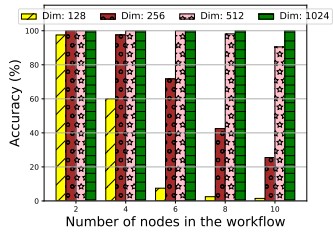

(a) Exp1: Random node sequences     (b) Exp2: Random walk, Wikispeedia     (c) Exp3: Random walk, Citation Network

Figure 5: Accuracy of decoding a node sequence vector (Dim: dimension of the vectors).

techniques. Second, even in cases where the difference in accuracy using the two different sources of information is large, SENSE-S (add) and (cat) are robust, and can effectively extract useful information from text descriptions to further reduce the error of link prediction that uses graph structure alone. Finally, this is significant because it demonstrates the effectiveness of SENSE-S in a variety of scenarios, including cases where the two sources of information may not be equally valuable.

## 5.4 SENSE EVALUATION

We now evaluate the accuracy of encoding/decoding by SENSE for node sequences. We evaluate on three experiments via constructing node sequences in the following different ways: **Experiment 1:** the node at every position in a sequence is chosen uniformly at random from $4,604$ Wikispeedia nodes. Note that as mentioned earlier, this node sequence may contain repeated nodes (which is allowed), and may or may not be a subgraph of the original graph. **Experiment 2:** the node sequences are constructed by performing random walks on the Wikispeedia graph. **Experiment 3:** the node sequences are constructed by performing random walks on the Physics Citation Network. Note that in both Experiments 2 and 3, adjacent nodes in the node sequences will have *related vector embeddings*. Next, for such constructed node sequences, their vector representations are computed by SENSE-S. Given these sequence vectors, we then decode the node at each position. We evaluate the decoding accuracy under different sequence lengths and vector dimensions, as reported in Figure 5.

From Figure 5 (a), we make the following observations. First, when the node sequence length is small, all vector dimensions lead to almost $100\%$ decoding accuracy. Second, as node sequence length increases, the decoding accuracy declines sharply especially in cases where the node vector dimensions are relatively small, i.e., $128$ and $256$. This is because, by Theorem 2, correlations between the involved node vectors cause inevitable errors. Such error accumulates when the length of the node sequence is large. Nevertheless, with sufficiently large node vector dimension, i.e., $1024$, even long sequences can be decoded perfectly, and with $512$, we can decode a workflow of length 10 with over $90\%$ accuracy. Interestingly, Figures 5 (b) and (c) also show similar trends. This is significant, as it shows that *even if node sequences are constructed from correlated node vectors, i.e., picked from same graph neighborhood, the decoding still achieves high accuracy*. This is because, as shown in Theorem 2, after cyclic shifting, the resulting vectors are orthogonal with high probability when the vector dimension is large (even if these vectors are originally related). Finally, in Figures 5 (a) and (b) the decoding algorithm needs to find the best match from among $4,592$ nodes (Wikispeedia network). In contrast, in Figure 5 (c), it is much more challenging to find the match among $27,770$ nodes. Yet, we are able to decode with high accuracy with theoretical guarantees.

## 6 CONCLUSION

We presented SENSE that learns semantically enriched vector representations of graph node sequences. To achieve this, we first developed SENSE-S that learns single node embeddings via a multi-task learning formulation that jointly learns the co-occurrence probabilities of nodes within a graph and words within a node-associated document. We evaluated SENSE-S against state-of-the-art approaches that leverage both graph and text inputs and showed that SENSE-S improves multi-label classification accuracy in Wikispeedia dataset by up to $50\%$ and link prediction over Physics Citation network by up to $78\%$. We then developed SENSE that is able to employ provable schemes for vector composition to represent node sequences using the same dimension as the individual node vectors from SENSE-S. We demonstrated that the individual nodes within the sequence can be inferred with a high accuracy (close to $100\%$) from such composite SENSE vectors.

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

## A    LEMMA FOR THE PROOF OF THEOREM 2

**Lemma 3.** *If* $\mathbf{x}$ *and* $\mathbf{y}$ *are $N$-dimensional unit vectors with i.i.d. random vector elements, then* $\mathbb{E}[\mathbf{x} \cdot \mathbf{y}] = 0$ *and* $\mathrm{Var}[\mathbf{x} \cdot \mathbf{y}] = 1/N$.

*Proof.* Since both $\mathbf{x}$ and $\mathbf{y}$ are unit vectors, we have $\mathbf{x} \cdot \mathbf{y} = ||\mathbf{x}||_2 ||\mathbf{y}||_2 \cos\theta = \cos\theta$, where $\theta$ is the angle between $\mathbf{x}$ and $\mathbf{y}$. Since $\mathbf{x}$ and $\mathbf{y}$ are not correlated and both $\mathbf{x}$ and $\mathbf{y}$ are uniformly distributed across the sphere surface, $\theta$ is also uniformly distributed, and thus $\mathbb{E}[\mathbf{x} \cdot \mathbf{y}] = 0$.

As $\mathbf{x} \cdot \mathbf{y}$ is purely determined by the angle $\theta$ between $\mathbf{x}$ and $\mathbf{y}$, without loss of generality, we select $\mathbf{y} = [1, 0, \ldots, 0]^T$ and only consider $\mathbf{x} = [x_1, x_2, \ldots, x_N]$ to be a unit vector with i.i.d. random vector elements. Then, $\mathbf{x} \cdot \mathbf{y} = x_1$. Therefore, $\mathrm{Var}[\mathbf{x} \cdot \mathbf{y}] = \mathbb{E}[x_1^2] - \mathbb{E}^2[x_1] = \mathbb{E}[x_1^2]$. Since all vector elements in $\mathbf{x}$ are identically distributed, we have $\mathbb{E}[\mathbf{x} \cdot \mathbf{x}] = \mathbb{E}[\sum_{i=1}^N x_i^2] = \sum_{i=1}^N \mathbb{E}[x_i^2] = N \cdot \mathbb{E}[x_1^2] = 1$. Therefore, $\mathrm{Var}[\mathbf{x} \cdot \mathbf{y}] = \mathbb{E}[x_1^2] = 1/N$.  □

Based on Lemma 3, we know that if $\mathbf{x}$ and $\mathbf{y}$ are $N$-dimensional unit vectors with i.i.d. vector elements, then event $\mathbf{x} \cdot \mathbf{y} = 0$ happens almost surely, i.e., $\Pr[\mathbf{x} \cdot \mathbf{y} = 0] \approx 1$, when $N$ is large. Using this lemma, we are ready to prove Theorem 2.

## B    PROOF OF THEOREM 2

*Proof.* Based on the proof of Lemma 3, we know that $\mathbf{x} \cdot \mathbf{y}$ is only determined by the angle $\theta$ between them. Therefore, again, without loss of generality, let $\mathbf{y} = [1, 0, \ldots, 0]^T$ and $\mathbf{x} = [x_1, x_2, \ldots, x_N]$. Then $\mathbf{x} \cdot \mathbf{y} = x_1 = c$ and $\mathbf{x} \cdot \mathbf{y}^{(m)} = x_\gamma$, where $\gamma = 1 + (m \mod N)$. Then $\mathbb{E}[\mathbf{x} \cdot \mathbf{y}^{(m)} | \mathbf{x} \cdot \mathbf{y} = c] = \mathbb{E}[x_\gamma | \mathbf{x} \cdot \mathbf{y} = c]$. As $m \neq kN$ ($k \in \mathbb{Z}^+$), we have $\mathbf{y} \neq \mathbf{y}^{(m)}$. Moreover, $x_\gamma$ is uniformly distributed across the sphere surface under the condition that $x_1 = c$. Therefore, $\mathbb{E}[x_\gamma | \mathbf{x} \cdot \mathbf{y} = c] = 0$, i.e., $\mathbb{E}[\mathbf{x} \cdot \mathbf{y}^{(m)} | \mathbf{x} \cdot \mathbf{y} = c] = 0$. Next, $\mathrm{Var}[\mathbf{x} \cdot \mathbf{y}^{(m)} | \mathbf{x} \cdot \mathbf{y} = c] = \mathbb{E}[x_\gamma^2 | x_1 = c] - \mathbb{E}^2[x_\gamma | x_1 = c] = \mathbb{E}[x_\gamma^2 | x_1 = c]$. Since $x_2, x_3, \ldots, x_N$ in $\mathbf{x}$ are identically distributed, we have $\mathbb{E}[\mathbf{x} \cdot \mathbf{x}] = \mathbb{E}[\sum_{i=1}^N x_i^2 | x_1 = c] = \sum_{i=1}^N \mathbb{E}[x_i^2 | x_1 = c] = c^2 + (N - 1)\mathbb{E}[x_\gamma^2 | x_1 = c] = 1$. Therefore, $\mathrm{Var}[\mathbf{x} \cdot \mathbf{y}^{(m)} | \mathbf{x} \cdot \mathbf{y} = c] = (1 - c^2)/(N - 1)$.  □

Based on Theorem 2, if $\mathbf{x}$ and $\mathbf{y}$ are $N$-dimensional unit vectors with i.i.d. vector elements, then given $\mathbf{x} \cdot \mathbf{y} = c$ ($|c| \leq 1$), event $\mathbf{x} \cdot \mathbf{y}^{(m)} = 0$ ($m \neq kN$ and $m, k \in \mathbb{Z}^+$) happens almost surely, i.e., $\Pr\left[\mathbf{x} \cdot \mathbf{y}^{(m)} = 0 | \mathbf{x} \cdot \mathbf{y} = c\right] \approx 1$, when $N$ is large.

