# OpenReview forum: "SENSE: SEMANTICALLY ENHANCED NODE SEQUENCE EMBEDDING"
_ICLR.cc/2019/Conference_

### Official Review · AnonReviewer1 · 2018-10-13
**Interesting idea and fleshed-out experiments, but somewhat niche appeal.**

**Rating:** 5
**Confidence:** 3

**Review:**

The paper proposes node embedding methods for applications where nodes are sequentially related. An example application is the "Wikispeedia" dataset, in which nodes are connected in a graph, but a datapoint (a wikispeedia "game") consists of a sequence of nodes that are visited. Each node is further attributed with textual information.

The methods proposed are most closely related to skipgrams, whereby the sequence of nodes are treated like words in a sentence. Then, node attributes (i.e., text) and node representations must be capable of predicting neighboring nodes/words. (Fig.s 1/2 are a pretty concise overview of the proposed architecture).

Positively, this is a quite sensible extension and modification of existing ideas in order to support a new (or different) problem setting.

Negatively, I'd say the applications for this technique are fairly niche, which may limit the paper's readership. The method is mostly fairly straightforward and not methodologically groundbreaking (probably borderline in terms of expected methodological contribution for ICLR). I also didn't understand whether the theoretical claims were significant.

The wikispedia/physics experiments feel a bit more like proofs-of-concept rather than demonstrating that the technique has compelling real-world uses. The experiments are quite well fleshed-out and detailed though.

---

> ### Author Response · Authors · 2018-11-26
> **motivation**
>
> Thanks a lot for the feedback. We just would like to highlight a couple of things:
>
> (I) Applications of SENSE-S: SENSE-S computes embeddings of single nodes using both graph structure and node features (text). In occasions where these both are important, SENSE-S is very applicable. For instance,
> (a) Recommendation systems frequently use node embeddings these days on “user-product” interaction (bipartite) graphs. This helps understand which products should be recommended to which users. Quite often in this situation, bootstrapping new products is a problem since no users have viewed or bought them. In this case, using their textual descriptions (along with graph structure) will help us obtain reasonable initial embeddings.
> (b) Link prediction in social networks is important to suggest new friends. This would depend on current friends of users (graph structure) as well as profile of users (textual descriptions). This is similar to the recommendation problem above, albeit in a different context.
>
> (II) Applications of SENSE: SENSE computes embeddings of node sequences with the same dimension as that of individual nodes. This is useful in a variety of scenarios, across different fields of computer science, where we want to represent a set of nodes while preserving the order. Representative applications include:
> (a) Source routing: This refers to a routing strategy in Internet, where the sender of a packet specifies the path that this packet takes through the network. The path specifies a certain order that needs to be preserved and SENSE can effectively do so.
> (b) Service composition: Recently microservices are getting popular, where a service is composed of several smaller microservices. Here, the order of execution becomes important and SENSE represents these complex services in the form of vectors to enable effective learning on vector representations.
> (c) Reading order of pages: For instance, reading pages in Wikipedia.
> (d) Representing any path in a graph or a network: For instance, representing the shortest/least congested/load balanced path between a pair of nodes.

---

### Official Review · AnonReviewer3 · 2018-11-03
**This paper presents two approaches: one called SENSE-S for embedding nodes in attributed networks; the other one called SENSE for embedding a sequence of nodes. SENSE-S follows the structure of Skip-gram model. The main difference is that SENSE-S considers both node and words in node content as input and output for learning their embedding.**

**Rating:** 4
**Confidence:** 5

**Review:**

This paper presents two approaches: one called SENSE-S for embedding nodes in attributed networks; the other one called SENSE for embedding a sequence of nodes. SENSE-S follows the structure of Skip-gram model. The main difference is that SENSE-S considers both node and words in node content as input and output for learning their embedding. For generating embedding vector for a sequence of nodes, SENSE takes summation of cyclically shifted unit-vectors constructed by SENSE-S on nodes in a sequence.

The paper is well written with a clear definition of the studied problem and a clear introduction of the presented methods. Evaluation was conducted on two real-world data sets (Wikipedia and citation network). It is an interesting idea to represent a sequence by the summation of cyclically shifted unit-vectors of nodes in a sequence. However, there are several concerns about the work presented in this paper.
1) the evaluation of SENSE-S is not sufficient. The baseline methods used in comparison are the simple ones that take concatenation of vectors induced from text and graph, or use one for initializing the learning of the other.  There existing several approaches that learn node embedding vectors from attributed graph (considering both the node content text and graph topology structure), such as TADW [1], HSCA [2], PLANE [3],GAE[4], AANE[5], ANRL [6]. SENSE-S should be compared with these methods for showing its effectiveness.
2) the embedding vector of a node sequence is evaluated by showing the decoding accuracy. It would be more interesting to show how these vectors can be used for some real applications. And, to have high decoding accuracy, the embedding dimension for sequences of 10 nodes should be up to 1024, which is quite expensive for computing and for storage, making the presented method unpractical in real-world applications.


[1] C. Yang, Z. Liu, D. Zhao, M. Sun, E. Y. Chang, Network representation learning with rich text information. IJCAI, 2015
[2] D. Zhang, J. Yin, X. Zhu, C. ZHang, Homophily, structure, and content augmented network representation learning.  ICDM 2016.
[3] T. M. V. Le and H. W. Lauw. Probabilistic latent document network embedding.  ICDM, 2014.
[4] Thomas N Kipf, Max Welling. Variational Graph Auto-Encoders. NIPS Workshop on Bayesian Deep Learning.  2016
[5] Xiao Huang, Jundong Li, Xia Hu. Accelerated attributed network embedding. SDM 2017.
[6] Zhen Zhang, Hongxia Yang, Jiajun Bu, Sheng Zhou, Pinggang Yu, Jianwei Zhang, Martin Ester, Can Wang. ANRL: Attributed Network Representation Learning via Deep Neural Networks. IJCAI, 2018

---

> ### Author Response · Authors · 2018-11-26
> **Thanks a lot!**
>
> We thank the reviewer for the constructive feedback! To the best of our knowledge SENSE-S is different from
> (i)             TADW (Network representation learning with rich text information), because TADW is based on DeepWalk, whereas SENSE-S has the flexibility that node2vec has in incorporating different ways of sampling node neighborhood to give different weightage to graph properties like homophily and structural equivalence as required. Moreover, SENSE-S is able to easily trade-off between the importance of graph weightage versus text weightage.  In addition, TADW uses TFIDF matrix to incorporate text information. In contrast, since we use skip-gram model, we can also account for the context in which different words are used within a document.
> (ii)            HSCA (Homophily, structure, and content augmented network representation learning), because HSCA just builds on TADW and adds additional term to ensure learning homophily. However, SENSE-S is able to easily trade-off between the importance of graph weightage versus text weightage.  Also, like TADW, HSCA also uses TFIDF matrix to incorporate text information.
> (iii)           PLANE (Probabilistic latent document network embedding), because the objective of PLANE is to maximize the likelihood that neighboring nodes have similar embeddings, which is not always the case in practice because neighboring nodes may be semantically different; more critically, it relies on strong assumptions of statistical distributions of words and edges in the network.
> (iv) VGAE (Variational Graph Auto-Encoders), is sensitive to the input node feature matrix X, as they do not provide a specific way to construct X. Further, SENSE-S is flexible enough to trade-off the relative importance of graph and node features.
> (v)           AANE (Accelerated attributed network embedding), because unlike SENSE-S, AANE does not account for the context in which different words are used within a document. Attributes are either keywords of blogs, tags from images in Flickr or bag-of-words representation of reviews in Yelp.
> (vi)          ANRL (ANRL: Attributed Network Representation Learning via Deep Neural Networks), because just like AANE, ANRL does not account for the context in which different words are used within a document.
> Also please note that even with 512 dimension node embedding, we are able to decode workflow of length 10 nodes with over 90% accuracy.

---

### Official Review · AnonReviewer2 · 2018-11-09
**Interesting topics are introduced but some corrections and clarifications are necessary**

**Rating:** 4
**Confidence:** 4

**Review:**

The authors introduce the problem of learning embeddings that consider both text information and graph structures, as well as the embedding of a sequence of nodes with embeddings.

However, the proposed algorithm, SENSE-S, is incremental in the sense of aggregating two simple structures. In the evaluation, it is compared only with the heuristic combination of node2vec and paragraph2vec, not with any existing work about the graph embeddings that incorporate node features even though they are mentioned in the related work.

Furthermore, the objective of node sequence embedding is not clear. What do we want to represent from the embedding of node sequences? It looks like we have to keep the node embeddings anyway, and then what is the problem of just storing node ordering instead of having representation? Or can we aggregate node embeddings in some way with storing the order of nodes? These kinds of questions can be raised, mainly because of uncertain objectives. The description of preserving both ordering and node properties is too vague.

Also, SENSE does not seem to have any connection with SENSE-S. Why is SENSE-S special to SENSE? Are they independent?

Finally, the authors claim that SENSE is necessary to overcome the space issue that needs q*d dimension. However, from Figure 5, it seems that the proposed algorithm actually needs O(q*d) dimensions to represent the sequence correctly. It is somewhat related to the question about i.i.d. assumption in Theorem 2, where embedding does not guarantee the orthogonality across the dimensions.

* Details
- In the introduction, "first" is repeated in the last paragraph of Page 1.
- N_G(v) and N_T(\phi) are said to be independent, but it should be the assumption since they are not the fact.
- Eq. (2) is not aligned with Eq (3) or (4). Either one needs to be fixed or the derivation needs to be described.
- How SVM is used needs to be described. Usage of embedding might be different depending on the usage of RBF kernel or linear kernel.
- Using the smaller number of random walks for Citation Network because it is a larger dataset needs some explanation.
- The calculation on the improvement percentage is completely misleading. If the accuracy is improved from 95% to 96%, it is about 1% improvement, not 20% improvement based on the error rate calculation.
- How the training/validation/test sets are split needs more description. Is it split by nodes or edges?

---

> ### Author Response · Authors · 2018-11-26
> **Thanks a lot for your feedback!**
>
> Objective of embedding sequences: Computes embeddings of node sequences with same dimension as that of individual nodes. This is useful in a variety of scenarios, across different fields of computer science, where we want to represent a set of nodes while preserving the order. Representative applications include:
> (a) Source routing: This refers to a routing strategy in Internet, where the sender of a packet specifies the path that this packet takes through the network. The path specifies a certain order that needs to be preserved and SENSE can effectively do so.
> (b) Service composition: Recently microservices are getting popular, where a service is composed of several smaller microservices. Here, the order of execution becomes important and SENSE represents these complex services in the form of vectors to enable effective learning on vector representations.
> (c) Reading order of pages: For instance, reading pages in Wikipedia.
> (d) Representing any path in a graph or a network: For instance, representing the shortest/least congested/load balanced path between a pair of nodes.
>
> Although we have to keep individual node embeddings, when workflow embeddings have to be communicated over a network, representing it with the same dimension as that of individual node vectors, as opposed to a list of node vectors, is helpful in reducing the total size of the transmitted information, while also providing secured information delivery because without the individual node embeddings, potential attackers are not able to decode a workflow.
>
> SENSE can also work on embeddings obtained via any other techniques (not just SENSE-S).
>
> SVM uses linear kernel.
>
> Train/Valid/Test split -- for node classification, it is by nodes; for link prediction, it is by links.

---

### Meta-Review · Area_Chair1 · 2018-12-20
**Not enough novelty and in a somewhat niche area**

**Confidence:** 5
**Recommendation:** Reject

**Metareview:**

The paper can also improved thorough a more thorough evaluation.